# Unsupervised Object Keypoint Learning using Local Spatial Predictability

**Anand Gopalakrishnan, Sjoerd van Steenkiste, Jürgen Schmidhuber**
The Swiss AI Lab IDSIA, USI, SUPSI
`{anand, sjoerd, juergen}@idsia.ch`

## Abstract

We propose *PermaKey*, a novel approach to representation learning based on object keypoints. It leverages the predictability of local image regions from spatial neighborhoods to identify salient regions that correspond to object parts, which are then converted to keypoints. Unlike prior approaches, it utilizes predictability to discover object keypoints, an intrinsic property of objects. This ensures that it does not overly bias keypoints to focus on characteristics that are not unique to objects, such as movement, shape, colour etc. We demonstrate the efficacy of *PermaKey* on Atari where it learns keypoints corresponding to the most salient object parts and is robust to certain visual distractors. Further, on downstream RL tasks in the Atari domain we demonstrate how agents equipped with our keypoints outperform those using competing alternatives, even on challenging environments with moving backgrounds or distractor objects.

## 1 Introduction

An intelligent agent situated in the visual world critically depends on a suitable representation of its incoming sensory information. For example, a representation that captures only information about relevant aspects of the world makes it easier to learn downstream tasks efficiently (Barlow, 1989; Bengio et al., 2013). Similarly, when explicitly distinguishing abstract concepts, such as objects, at a representational level, it is easier to generalize (systematically) to novel scenes that are composed of these same abstract building blocks (Lake et al., 2017; van Steenkiste et al., 2019; Greff et al., 2020).

In recent work, several methods have been proposed to learn unsupervised representations of images that aim to facilitate agents in this way (Veerapaneni et al., 2019; Janner et al., 2019). Of particular interest are methods based on learned *object keypoints* that correspond to highly informative (salient) regions in the image as indicated by the presence of object parts (Zhang et al., 2018; Jakab et al., 2018; Kulkarni et al., 2019; Minderer et al., 2019). Many real world tasks primarily revolve around (physical) interactions between objects and agents. Therefore it is expected that a representation based on a set of task-agnostic object keypoints can be re-purposed to facilitate downstream learning (and generalization) on many different tasks (Lake et al., 2017).

One of the main challenges for learning representations based on object keypoints is to discover salient regions belonging to objects in an image without supervision. Recent methods take an information bottleneck approach, where a neural network is trained to allocate a fixed number of keypoints (and learn corresponding representations) in a way that helps making predictions about an image that has undergone some transformation (Jakab et al., 2018; Minderer et al., 2019; Kulkarni et al., 2019). However, keypoints that are discovered in this way strongly depend on the specific transformation that is considered and therefore lack generality. For example, as we will confirm in our experiments, the recent *Transporter* (Kulkarni et al., 2019) learns to prioritize image regions that change over time, even when they are otherwise uninformative. Indeed, when relying on *extrinsic* object properties (i.e. that are not unique to objects) one becomes highly susceptible to distractors as we will demonstrate.

In this work, we propose *PermaKey* a novel representation learning approach based on object keypoints that does not overly bias keypoints in this way. The key idea underlying our approach is to view objects as local regions in the image that have high internal predictive structure (self-information). We argue that local predictability is an intrinsic property of an object and therefore more reliably captures *objectness* in images (Alexe et al., 2010). This allows us to formulate a *local spatial prediction*

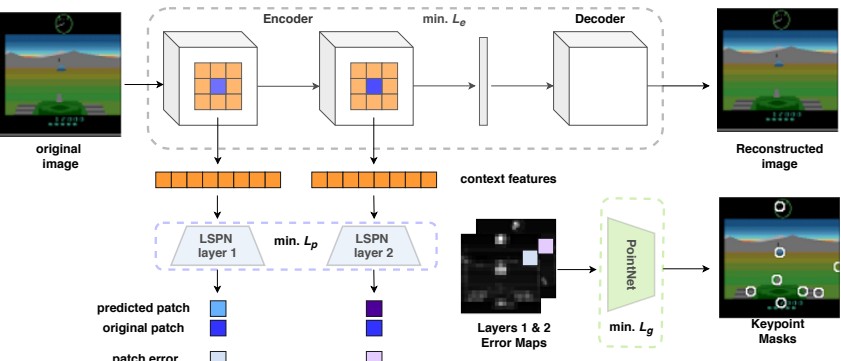

Figure 1: *PermaKey* consists of three modules (encapsulated by the dotted lines): learning a suitable spatial feature embedding (1), solving a local spatial prediction task (2), and converting error maps to keypoints (3). Objective functions used to learn each of the 3 modules shown within dotted blocks.

*problem* to infer which of the image regions contain object parts. We perform this prediction task in the learned feature space of a convolutional neural network (CNN) to assess predictability based on a rich collection of learned low-level features. Using *PointNet* (Jakab et al., 2018) we can then convert these predictability maps to highly informative object keypoints.

We extensively evaluate our approach on a number of Atari environments and compare to *Transporter* (Kulkarni et al., 2019). We demonstrate how our method is able to discover keypoints that focus on image regions that are unpredictable and which often correspond to salient object parts. By leveraging local predictability to learn about objects, our method profits from a simpler yet better generalizable definition of an object. Indeed, we demonstrate how it learns keypoints that do not solely focus on temporal motion (or any other extrinsic object property) and is more robust to uninformative (but predictable) distractors in the environment, such as moving background. On Atari games, agents equipped with our keypoints outperform those using *Transporter* keypoints. Our method shows good performance even on challenging environments such as *Battlezone* involving shifting viewpoints where the *Transporters'* explicit motion bias fails to capture any task-relevant objects. As a final contribution, we investigate the use of graph neural networks (Battaglia et al., 2018) for processing keypoints, which potentially better accommodates their discrete nature when reasoning about their interactions, and provide an ablation study.

## 2 METHOD

To learn representations based on task-agnostic object keypoints we require a suitable definition of an object that can be applied in an unsupervised manner. At a high level, we define objects as abstract patterns in the visual input that can serve as *modular* building blocks (i.e. they are self-contained and reusable independent of context) for solving a particular task, in the sense that they can be separately intervened upon or reasoned with (Greff et al., 2020). This lets us treat objects as *local regions* in input space that have high internal predictive structure based on statistical co-occurrence of features such as color, shape, etc. across a large number of samples. Hence, our focus is on their local predictability, which can be viewed as an "intrinsic" object property according to this definition. For example, Bruce & Tsotsos (2005) have previously shown that local regions with high self-information typically correspond to salient objects. More generally, self-information approximated via a set of cues involving center-surround feature differences has been used to quantify *objectness* (Alexe et al., 2010).

In this paper we introduce *Prediction ERror MAp based KEYpoints (PermaKey)*, which leverages this definition to learn about object keypoints and corresponding representations. The main component of *PermaKey* is a *local spatial prediction network* (LSPN), which is trained to solve a local spatial prediction problem in feature space (light-blue trapezoid in Figure 1). It involves predicting the value of a feature from its surrounding neighbours, which can only be solved accurately when they belong to the same object. Hence, the error map (predictability map) that is obtained by evaluating the LSPN at different locations carves the feature space up into regions that have high internal predictive structure (see rows 4 & 5 in Figure 2(a)). In what follows, we delve into each of the 3 modules that constitute

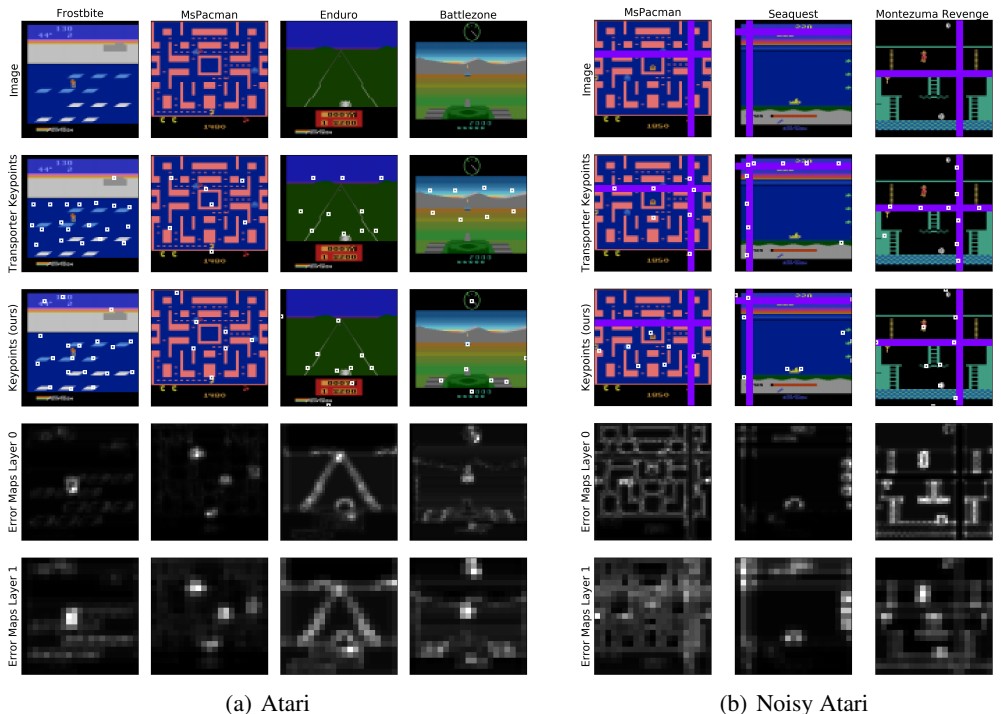

(a) Atari  (b) Noisy Atari

Figure 2: Image (row 1), *Transporter* keypoints overlaid on image (row 2), *PermaKey* (ours) keypoints overlaid on image (row 3), predictability maps from feature layers 0 and 1 (rows 4 & 5 respectively).

our *PermaKey* system: 1) the VAE to learn spatial features 2) the LSPN to solve the local spatial prediction task and 3) PointNet to group error maps into keypoints (see Figure 1 for an overview).

**Spatial Feature Embedding**   To solve the local spatial prediction task we require an informative set of features at each image location. While prior approaches focus on low-level features of image patches (eg. RGB values Isola et al. (2014), or ICA features  Bruce & Tsotsos (2005)), we propose to learn features using a Variational Auto-Encoder (VAE; Kingma & Welling (2014)). It consists of an *Encoder* that parametrizes the approximate posterior $q_\phi(z|x)$ and a *Decoder* that parametrizes the generative model $p_\theta(x_i|z)$, which are trained to model the observed data via the ELBO objective:

$$L_e(\theta, \phi) = \mathbb{E}_{q_\phi(z|x)} \log[p_\theta(x|z)] - D_{KL}[q_\phi(z|x)||p(z)]. \tag{1}$$

The encoder is based on several layers of convolutions that offer progressively more abstract image features by trading-off spatial resolution with depth. Hence, which layer(s) we choose as our feature embedding will have an effect on the outcome of the local spatial prediction problem. While more abstract high-level features are expected to better capture the internal predictive structure of an object, it will be more difficult to attribute the error of the prediction network to the exact image location. On the other hand, while more low-level features can be localized more accurately, they may lack the expressiveness to capture high-level properties of objects. Nonetheless, in practice we find that a spatial feature embedding based on earlier layers of the encoder works well (see also Section 5.3 for an ablation).

**Local Spatial Prediction Task**   Using the learned spatial feature embedding we seek out salient regions of the input image that correspond to object parts. Our approach is based on the idea that objects correspond to local regions in feature space that have high internal predictive structure, which allows us to formulate the following local spatial prediction (LSP) task. For each location in the learned spatial feature embedding, we seek to predict the value of the features (across the feature maps) from its neighbouring feature values. When neighbouring areas correspond to the same object-(part), i.e. they regularly appear together, we expect that this prediction problem is easy (green arrow in Figure 3). In contrast, this is much harder when insufficient context is available (red arrow

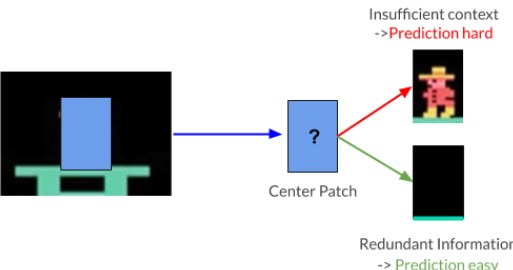

Figure 3: Two different prediction scenarios encountered when attempting to solve the LSP task. In the first case, too little context information is available to predict the center patch from its surroundings, likely yielding a prediction error (assuming the character is not *always* present). Alternatively, when sufficient context is available as in the second case, prediction becomes easy.

in Figure 3), or when features rarely co-occur. Similarly, when parts of an image remain fixed across a large number observations it is easily predictable regardless of how much context is available.

We implement the local spatial prediction task by training a neural network (LSPN in Figure 1) to predict a center patch of features from the 8 first-order neighbouring patches using the learned spatial feature embedding at layer $l$ of the VAE encoder. The weights of this network are trained to solve this prediction problem across images and across feature locations using the following objective:

$$L_p(\psi_l) = \frac{1}{NHW} \sum_x^{H,W} \sum_{i,j} (f_{\psi_l}^{\text{LSPN}}(A[ne(i,j)]) - A[i,j])^2, \quad (2)$$

where $A \in \mathbb{R}^{H \times W \times C}$ is the set of feature maps at layer $l$ of the encoder $q_\phi(z|x)$, and $ne(i,j)$ are the coordinates of the first-order neighbours surrounding $i, j$ and $H, W$ are the height and width of the spatial feature maps respectively. The features of the neighbouring locations are concatenated before being fed to the network. We train a separate LSPN for each of the chosen feature embedding specified by the encoder layer(s) and record their error at each location to obtain *predictability maps* that are expected to indicate the presence of objects (rows 4 & 5 in Figure 2(a)).

**Extracting Keypoints from Error Maps** The predictability maps can be thought of as being a superposition of the local predictability errors due to each of the individual objects in the scene. Hence, what remains is to extract the $k$ most salient regions (belonging to object parts) as our object keypoints from these predictability maps. We do so by training a *PointNet* (Jakab et al., 2018) to reconstruct the error maps through a bottleneck consisting of $k$ fixed-width 2D Gaussian windows with learnable centers $\mu$ that are the keypoints ($L_g$ loss in Figure 1) as shown below:

$$L_g(\omega) = \frac{1}{MHW} \sum_m^{M} \sum_{i,j}^{H,W} (f_\omega^{\text{PointNet}}(\mathcal{H}[m,i,j]) - \mathcal{H}[m,i,j])^2. \quad (3)$$

Here $\mathcal{H} \in \mathbb{R}^{M \times H \times W}$ is the concatenation of layer-wise LSP error maps for various encoder layer(s) $l$ ($M$ in total), and $H, W$ are the height and width of the error maps (potentially up-sampled). Since the bottleneck layer has only limited capacity it is forced to consider locations that are most helpful in minimizing reconstruction error. This also requires the *PointNet* to consider statistical regularities across the predictability maps that can efficiently be described. In that sense this procedure can be thought of as a way of (spatially) clustering the local errors belonging to the same object by placing keypoints.

## 3 USING KEYPOINT REPRESENTATIONS FOR DOWNSTREAM PROCESSING

Given a set of keypoint locations and their associated features learned in an purely unsupervised framework we examine how to incorporate them into a state representation suitable to solve a set of downstream RL tasks. Here we will consider Atari games as an example, which essentially consist of entities such as a player, enemies, reward objects like health potions, food etc. Player(s) obtain positive reward through collecting these reward objects or killing enemies and in turn risk getting attacked by these enemies. Successful game-play in such environments would therefore require an understanding of these interaction effects between player, enemies and reward objects in order to estimate the best next set of actions to take to maximize the game score.

Prior work like the *Transporter* (Kulkarni et al., 2019) uses a CNN to encode features of object keypoints for downstream RL tasks on the Atari domain. However, a CNN permits an implicit encoding of relational factors between keypoints only in a limited sense since it measures interactions between spatially close keypoints through the weight-sharing scheme of the convolutional kernel. An alternative is to use a graph neural network (GNN) (Scarselli et al., 2009; Battaglia et al., 2018) (see also Pollack (1990); Küchler & Goller (1996) for earlier approaches) which have been applied to model relational inductive biases in the state representations of deep RL agents (Zambaldi et al., 2019). In a similar spirit, we consider treating keypoints as an unordered set and using a GNN to explicitly encode relational factors between pairs of such entities. In this case, we spatially average the convolutional features within each keypoint mask to obtain a keypoint feature. These are appended with learned positional embedding and initialized as node values of the underlying graphs. Underlying graphs are fully-connected and edge factors initialized with zeros. We use the *Encode-Process-Decode* design strategy (Battaglia et al., 2018) for the graph keypoint encoder in our PKey+GNN model for Atari in Section 5.1. The *Encode* and *Decode* blocks independently process nodes and edges of underlying graphs. We use an *Interaction Network* (Battaglia et al., 2016) for the *Process* block with a single message-passing step to compute updated edges $e'$ and nodes $v'$ as shown below:

$$e'_{ij} = f^e([v_i, v_j, e_{ij}]), \qquad \overline{e}'_i = \rho^{e \to v}(E'_i), \qquad v'_i = f^v([v_i, \overline{e}'_i]). \tag{4}$$

Here $e_{ij}$ is the edge connecting nodes $v_i$ and $v_j$, $f^e$ is the edge update function, $\rho^{e \to v}$ is the edge aggregation function, $E'_i$ represents all incoming edges to node $v_i$ and $f^v$ is the node update function. After one round of message-passing by the *Interaction Network* the resultant nodes are decoded as outputs. These output node features of all nodes are then concatenated and passed through an MLP before being input to the agent. For further implementation details we refer to Appendix A.2.3.

## 4 RELATED WORK

Inferring object keypoints offers a flexible approach to segmenting a scene into a set of informative parts useful for downstream processing. Recent approaches (Jakab et al., 2018; Zhang et al., 2018; Kulkarni et al., 2019; Minderer et al., 2019) apply a transformation (i.e. rotation, deformation, viewpoint shift or temporal-shift) on a source image to generate a target image. An autoencoder network with a keypoint-bottleneck layer is trained to reconstruct the target image given the source image and thereby learns to infer the most salient image regions that differ between source and target image pairs (Jakab et al., 2018). However, since the initial transformation does not focus on intrinsic object properties, they are more susceptible to 'false-positives' and 'false-negatives'. In extreme scenarios, such approaches may even fail to infer any useful object keypoints at all (e.g. as shown in Figure 2(b)).

The more general framework of discovering salient image regions (Itti et al., 1998) is closely related to inferring object keypoints. Early work explored the connections between information theoretic measures of information gain or local entropy (Kadir & Brady, 2001; Jagersand, 1995) and its role in driving fixation(attention) patterns in human vision. For example, Bruce & Tsotsos (2005) use Shannon's self information metric as a measure of saliency, while Zhmoginov et al. (2019) leverage mutual information to guide saliency masks. An interesting distinction is between *local* and *global* saliency, where the former is concerned with (dis)similarity between local image regions and the latter focuses on how (in)frequently information content occurs across an entire dataset (Borji & Itti, 2012). Our prediction objective accounts for both: the LSP account for local saliency, while more global saliency is achieved having the prediction network solve this local prediction problem across an entire dataset of images. In this way, it can also reason about statistical regularities that only occur at this more global level.

An alternative to inferring object keypoints (and corresponding representations) is to learn entire object representations. Recent approaches can be broadly categorized into spatial mixture models (Greff et al., 2017; 2019; Burgess et al., 2019), sequential attention models (Eslami et al., 2016; Kosiorek et al., 2018) or hybrid models that are combinations of both (Lin et al., 2020). While these methods have shown promising results, they have yet to scale to more complex visual settings. In contrast, object keypoints focus on more low-level information (i.e. object parts) that can be inferred more easily, even in complex visual settings (Minderer et al., 2019) Recent work on object detection (Duan et al., 2019; Zhou et al., 2019) have explored the use of a center point (i.e. keypoint) and a bounding box as an object abstraction. However, they use ground-truth bounding boxes of objects to infer keypoint locations whereas our method is fully unsupervised. Alternatively Isola et al.

(2015) propose a self-supervised framework for instance segmentation where local image patches are classified as belonging together based on their spatial or temporal distance.

Graph neural networks (GNNs) have been previously used for intuitive physics models (Battaglia et al., 2016; Chang et al., 2017; van Steenkiste et al., 2018; Janner et al., 2019; Veerapaneni et al., 2019; Kipf et al., 2020, Stanić et al., 2021). Other work instantiate the policy (Wang et al., 2018) or value function (Bapst et al., 2019) as GNNs for RL tasks on continuous control or physical construction domains. However, the latter assume access to underlying ground-truth object(-part) information. In contrast, we learn our keypoint representations in a fully unsupervised manner from raw images. Further, we do not instantiate the policy or value function as a GNN, but rather use it to model relational factors between learned keypoint for downstream RL tasks in a similar spirit to Zambaldi et al. (2019).

## 5 EXPERIMENTS

We empirically evaluate the efficacy of our approach on a subset of Atari games (Bellemare et al., 2013) that ensure a good variability in terms of object sizes, colour, types, counts, and movement. As a baseline, we compare against the recently proposed *Transporter* (Kulkarni et al., 2019), which was previously evaluated on Atari. To evaluate the efficacy of object keypoints to facilitate sample-efficient learning on RL tasks we train agents on the low-data regime of 100,000 interactions following the same experimental protocol as in Kulkarni et al. (2019); Kaiser et al. (2020).

We use a deep recurrent Q-learning variant (Hausknecht & Stone, 2015) using double-Q learning (Hasselt et al., 2016), target networks (Mnih et al., 2015) and a 3-step return as our RL algorithm for all agents with a window size of 8 and batch size of 16. To implement the Q-function we use a 128 unit LSTM network (Hochreiter & Schmidhuber, 1997; Gers, 1999). Full experimental details can be found in Appendix A and additional results in Appendix B[1].

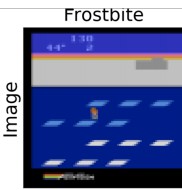

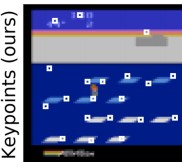

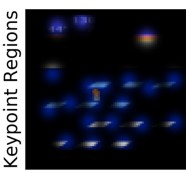

### 5.1 ATARI

We train *PermaKey* on Atari and find that it is frequently able to recover keypoints corresponding to object parts. For example, in Figure 2(a) (row 3) it can be seen how our method discovers keypoints correponding to the road and traffic in *Enduro* (column 3) and player and enemy tank parts (eg. wheels, cannon) in *Battlezone* (column 4). On *Frostbite* we find that it learns about the borders of the ice platforms, the player, scoreboard and birds, while on *MsPacman* we observe that it tracks the various characters like the blue ghosts and Pac-Man. We emphasize that while the precise location of the keypoints often seems to focus on the *borders* of predictability, their associated window of attention (i.e. in *PointNet*) is able to capture most of the object part. This is better seen in Figure 4 (and in Figure 10 in Appendix B), where the corresponding windows of attention (available to an agent) are shown. From the error maps (rows 4 and 5 in Figure 2(a)) it can be seen how the LSPN implicitly learns to discover salient parts of the image, and how the choice of feature embedding shifts the focus from local edges to entire objects. In Figure 5 (and Figures 8 and 9) it is shown how *PermaKey* is stable across multiple different runs.

Figure 4: Keypoint regions highlighted.

Compared to *Transporter* (Figure 2(a) row 2) we find that our approach often produces qualitatively better keypoints. For example, on *Enduro* and *Battlezone* the *Transporter* fails to place keypoints on salient object parts belonging to the road and traffic or parts of the player and enemy tank (eg. its canon, wheels etc.). This is intuitive, since in these environments moving image regions do not always correspond to objects. Indeed, on *Frostbite* and *MsPacman*, where this is not the case, the keypoints produced by *Transporter* and our method are of comparable quality.

In Table 1 we provide quantitative results of training agents that use these keypoints on 4 Atari environments. We observe that agents using *PermaKey* keypoints (PKey-CNN in Table 1) outperform corresponding variants using *Transporter* keypoints (original from Kulkarni et al. (2019) and our

---

[1]Code to reproduce all experiments is available at `https://github.com/agopal42/permakey`.

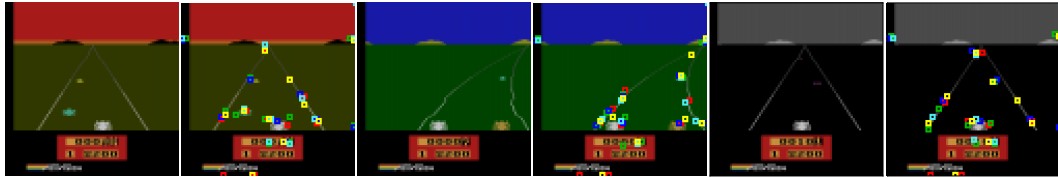

Figure 5: Observed *Enduro* frames (odd columns) and with overlaid keypoints from 5 random seeds having a unique color (even columns). The inferred keypoints are stable across different runs and consistently focus on salient object parts. Additional frames can be seen in Figure 8 in Appendix B.

| Game | Rainbow | PPO | SimPLE | Transporter (orig.) | Transporter (re-imp.) | Transporter -GNN | PKey-CNN (ours) | PKey-GNN (ours) |
|---|---|---|---|---|---|---|---|---|
| Battlezone | 3363.5 (523.8) | 5300.0 (3655.1) | 5184.4 (1347.5) | N/A | N/A | N/A | 10266.7 (3272.1) | **12566.7** (3297.9) |
| MsPacman | 364.3 (20.4) | 496.0 (379.8) | 762.8 (331.5) | 999.4 (145.4) | 983.0 (806.3) | 838.3 (537.3) | **1038.5** (417.1) | 906.3 (382.2) |
| Seaquest | 206.3 (17.1) | 370.0 (103.3) | 370.9 (128.2) | 236.7 (22.2) | 455.3 (160.6) | 296.0 (82.5) | **520.0** (159.9) | 375.3 (75.4) |
| Frostbite | 140.1 (2.7) | 174.0 (40.7) | 254.7 (4.9) | 388.3 (142.1) | 263.7 (14.9) | 465.0 (442.1) | 310.7 (150.1) | **657.3** (556.8) |

Table 1: Atari Results: mean scores (and std. dev.) obtained for each agent. For PKey and Transporter (re-imp.) we report the mean for 3 different policies (using the median evaluation run over 10 environment seeds). Results for Transporter (orig.) are taken from Kulkarni et al. (2019) and for SimPLE, PPO, and Rainbow from (Kaiser et al., 2020), where a similar experimental set-up is used. "N/A" indicates that a policy did not make any learning progress w.r.t a random policy.

re-implementation) on 3 environments[2]. Notice how on *Battlezone* we were unable to train a policy that is better than random when using *Transporter* keypoints. This is intuitive, since in Figure 2(a) we observed how *Transporter* learns keypoints that are not concerned with salient objects on this environment. For comparison, we also include scores from Kaiser et al. (2020) in Table 1, where a similar experimental setup is used. It can be seen that SimPLE (Kaiser et al., 2020), Rainbow (Hessel et al., 2018) and PPO (Schulman et al., 2017) also achieve worse scores than our PKey agents.

When using a GNN for downstream processing (PKey+GNN) we are able to obtain additional improvements on the *Battlezone* and *Frostbite* environments. Meanwhile on *MsPacman* and *Seaquest* performance drops, indicating that the stronger spatial inductive bias (i.e. nearby keypoints are processed similarly) implemented by CNNs can be advantageous as well. Note that the relative improvements of (Transporter+GNN) compared to (Transporter+CNN), largely mirrors the relative changes shown by analogous *PermaKey* variants (CNN vs GNN keypoint encoder) across all environments. Further, it can be seen how *PermaKey* consistently outperforms *Transporter*, even when using GNN keypoint encoders for both. On *MsPacman* we observed a larger standard-deviation for *Transporter*, compared to the results reported by Kulkarni et al. (2019), although we did tune the hyper-parameters for our re-implementation separately. A possible reason could be due to us using a lower batch size of 16 (compared to 32) and a lower window size of 8 (compared to 16). We were unable to investigate this further due to constraints on the available amount of compute. On Frostbite, we observe a large variances for both (PKey+GNN) and (Transporter+GNN) agents as some policies complete the first level (resulting in very high scores of about 2000 points), while others learn sub-optimal strategies yielding episodic returns around 220.

## 5.2 ROBUSTNESS TO DISTRACTORS

In order to demonstrate the limitations of the explicit motion bias in *Transporter* we synthesized a modified version of Atari ("noisy" Atari) that contains a coloured strip (either horizontal, vertical or both) at random locations in the environment (see row 1 in Figure 2(b)), thereby creating the illusion

---

[2]Unfortunately, Kulkarni et al. (2019) have not open-sourced their code (needed to reproduce RL results and apply *Transporter* on other environments) and their self-generated dataset containing 'ground-truth' keypoints for evaluation on Atari. We were able to obtain certain hyper-parameters through personal communication.

| Game | Transp. (re-imp.) | PKey + CNN |
|---|---|---|
| Seaquest | 357.3 (99.10) | **562.0** (114.78) |
| MsPacman | 470.0 (212.90) | **574.7** (290.07) |

Table 2: "Noisy" Atari Results: mean (and std. dev.) for 3 different policies (using the median over environment seeds as before).

| MsPacman | 5 keypoints | 7 keypoints | 10 keypoints |
|---|---|---|---|
| Transp. (re-imp.) | 923.0 (433.95) | 983.0 (806.3) | 907.0 (317.41) |
| PKey + CNN | **1004.3** (319.15) | **1038.5** (417.1) | **1003.3** (313.07) |

Table 3: Keypoint ablation on the *MsPacman* env.

of motion. Figure 2(b) (row 3) demonstrates how our method is more resistant against this distractor by focusing on predictability and is able to recover keypoints belonging to object parts as before (eg. player, skull, ladder positions in *Montezuma's Revenge*). Indeed, notice how the distractor does not show up in the error maps (rows 4 and 5 in Figure 2(b)) as it corresponds to a predictable pattern that was frequently encountered across the entire dataset[3]. In contrast, we find that *Transporter* (row 2) dedicates many of the available keypoints to the distractor as opposed to the static salient object parts.

As a quantitative experiment, we consider the downstream performance of agents that receive either *Transporter* keypoints or *PermaKey* keypoints on "Noisy" versions of *Seaquest* and *MsPacman*. In Table 2 it can be seen how the performance of the *PermaKey* agent (PKey + CNN) remains largely unaffected on *Seaquest* (compared to Table 1), while the sensitivity of *Transporter* to such moving distractors causes a significant drop in performance. On "Noisy" *MsPacman*, both agents perform worse compared to before, but the *PermaKey* only to a lesser extent. We hypothesize that the general performance drop is caused by the distractor bars occluding the fruits (collecting fruits yields rewards) located along the horizontal and vertical columns at every timestep. In general, we observe that *PermaKey* consistently outperforms *Transporter* on these tasks.

## 5.3 ABLATION STUDY

**Number of keypoints** We gain additional insight in our approach by modulating the number of keypoints and observe its qualitative effects. On the Frostbite environment we use 15, 20, 25 keypoints (shown in Figure 6), while keeping all the other hyperparameters same as described in Appendix A. When using additional keypoints we observe that *PermaKey* prioritizes regions in the error map that contribute most to the reconstruction loss. On the other hand, when the number of keypoints is too large, redundant keypoints are placed at seemingly random locations e.g. the border of the image.

We also quantitatively measure the effects of varying the number of keypoints $\{5, 7, 10\}$ produced by *PermaKey* and *Transporter* on downstream performance in *MsPacman* (Table 3), using the same evaluation protocol as before. The agent using *PermaKey* outperforms the one using *Transporter* keypoints across the entire keypoint range. Since the *Transporter* keypoints are biased towards image regions that change across frames, additional keypoints only seem to track the different characters that move around the maze. With *PermaKey* focusing on predictability, additional keypoints tend to capture other salient parts of the maze, although this did not yield improved downstream performance in this case.

**Spatial resolution of the feature embedding** We experiment with the choice of feature layer(s) used for the local spatial prediction task and observe its qualitative effects on the keypoints discovered. On the Space Invaders environment we use the following sets of feature layer(s) $= \{(0), (0, 1), (2, 3), (0, 1, 2, 3)\}$ and retain the same values for all the other hyperparameters as described in Appendix A. Results are shown in Figure 7 in Appendix B. When only the lower layers are used, we find that the corresponding error maps tend to focus more on highly local image features. In contrast, when only higher layers are used, we find that the loss of spatial resolution leads to highly coarse keypoint localization. In practice, we found that using feature layers 0 and 1 offers the right balance between spatial locality and expressiveness of features.

The ice floats in *Frostbite*, which are captured as keypoints (Figure 2(a)), serve as a useful example that illustrates the importance of integrating multiple spatial resolutions (i.e. that trade off local and global information as also noted in Itti et al. (1998); Jagersand (1995)). Ice floats seen at the default image resolution (layer-0 with 'conv' receptive field=4) might seem very predictable in it's highly local context of other ice float parts. However, casting the same local prediction task of predicting

---

[3]This observation also confirms that the LSPN does not collapse to a naive color thresholder.

the ice float features at layer-1 (image resolution 0.5x due to stride=2 in conv-encoder) given only blue background context would lead to some error spikes. This is because at many locations in the image, it seems reasonable to expect blue background patches to surround other blue background patches. Only at certain locations white ice floats are interspersed with blue background patches thereby driving some amount of error (surprise) in local predictability (In Figure 2(a) column 1, compare rows 4&5). Hence, varying the spatial resolution of the feature embedding and integrating the corresponding error maps lets us accommodate objects of varying shapes, sizes and geometry etc.

## 6 DISCUSSION

We have seen how *PermaKey*, by focusing on local *predictability*, is able to reliably discover salient object keypoints while being more robust to certain distractors compared to *Transporter* (Kulkarni et al., 2019). However, it should be noted that *PermaKey* can only account for a "bottom-up" notion of saliency (Itti, 2007) based on the observed inputs, and does not consider task-dependent information. Therefore a potential limitation of this approach is that it might capture objects that are not relevant for solving the task at hand (i.e. corresponding to "unpredictable distractors"). On the other hand, a purely unsupervised approach to extracting keypoints may allow for greater re-use and generalizability. The same (overcomplete) set of keypoints can now be used to faciliate a number of tasks, for example by querying (and attending to) only the relevant keypoints using top-down attention (Mott et al., 2019; Mittal et al., 2020). In this way it may readily facilitate other tasks, such as a "new object" acting as the key in *Montezuma's Revenge*, without having to re-train the bottom-up vision module.

Other kinds of "unpredictable distractors" could include certain types of image noise having highly unpredictable local structure, such as salt-and-pepper noise. It is unclear how this affects the local spatial prediction problem, since the learned spatial embedding of the VAE may readily suppress such information (due to it being trained for reconstruction). In that case, since *PermaKey* acts on these learned spatial embeddings, it could leave the quality of error map(s) relatively unchanged. Further, we note that such distractors are typically the result of artificial noise and uncommon in the real world, unlike distractors due to motion, which are therefore more important to handle effectively.

In this work, we have trained *PermaKey* in a purely unsupervised fashion by pre-training it once on a dataset of collected episodes of game-play. On games where new objects are introduced only at much later levels it would be beneficial to continue refining the keypoint module using the unsupervised loss (Kulkarni et al., 2019). Further, it may be interesting to consider whether there are benefits to learning both the keypoint module and policy network in an end-to-end fashion. While this is expected to reduce the re-use and generalizability of the discovered keypoints, it may make it easier to learn relevant keypoints due to directly considering task-specific information (e.g. via policy gradients).

## 7 CONCLUSION

We proposed *PermaKey*, a novel approach to learning object keypoints (and representations) that leverages *predictability* to identify salient regions that correspond to object parts. Through extensive experiments on Atari it was shown how our method is able to learn accurate object keypoints that are more robust to distractors, unlike the *Transporter* baseline approach. RL agents that employ *PermaKey* keypoints as input representations show sample-efficient learning on several Atari environments and outperform several other baselines. Further, it was shown how additional improvements can sometimes be obtained when processing keypoints using graph neural networks. Future work includes scaling *PermaKey* to more complex 3D visual worlds involving changing viewpoints, occlusions, egocentric vision etc. Similarly, designing quantitative evaluation metrics to directly measure the quality of keypoints extracted by different methods without the need for computationally expensive downstream tasks would allow for efficient performance analysis and model design as we begin to scale-up these ideas to more complex visual domains.

### ACKNOWLEDGMENTS

We thank Aleksandar Stanić and Aditya Ramesh for valuable discussions. This research was partially funded by ERC Advanced grant no: 742870 and by Swiss National Science Foundation grants: 200021‗165675/1 & 200021‗192356. We also thank NVIDIA Corporation for donating a DGX-1 as part of the Pioneers of AI Research Award and to IBM for donating a Minsky machine.

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

## A    Experiment Details

### A.1    Datasets

To obtain Atari game frames, we use rollouts of various pre-trained agents in the Atari Model Zoo (Such et al., 2019). We split the aggregated set of game frames for each of the chosen environments into separate train, validation and test sets of 85,000, 5000 and 5000 samples respectively.

For "noisy" Atari, we start with the regular Atari dataset and superimpose colored bars (either horizontal, vertical or both) centered at random x-y co-ordinates. This is done on-the-fly during training/evaluation to generate the required Noisy Atari samples.

### A.2    Architecture and Training Details

#### A.2.1    PermaKey

We train our method using the Adam optimizer (Kingma & Ba, 2015) with a initial learning rate of 0.0002 and decay rate of 0.85 every 10000 steps. We use a batch size of 32 and in all cases train for 100 epochs, using use early stopping of 10 epochs on the validation set to prevent overfitting. All modules (i.e. the VAE, the prediction network, and *PointNet*) are trained in parallel using their respective losses. We do not backpropagate gradients between modules to improve stability. The same hyperparameters are used across all environments except for the number of keypoints.

**Variational Autoencoder**    We use a convolutional neural network with 4 *Conv-BatchNorm-ReLU* layers for the encoder network of the VAE. The encoder network uses kernel sizes $[4, 3, 3, 3]$, filters $[32, 64, 64, 128]$ and strides $[1, 2, 2, 1]$. The architecture of the decoder is the transpose of that of the encoder with $2\times$ bi-linear upsampling used to undo the striding.

Preliminary experiments were used to determine the kernel size of the convolutional layers in the encoder. There it was observed that a kernel size of 4 on the first layer for $84 \times 84$ images achieves the best results and that larger kernel sizes and strides ($\geq 2$) reduce the spatial resolution of the feature maps and lead to coarser keypoint localization.

**Prediction Network**    We use a separate 3-layer MLP with hidden layer sizes $[8 \times p \times p \times C, 512, 256, p \times p \times C]$ where $p$ denotes height and width of activation patch and $C$ the number of channels of the activation map of encoder CNN. We use linear output activations for the prediction network and use a separate network for each of the selected layers of the VAE encoder that perform a separate local spatial prediction task. We use encoder layers $[0, 1]$ and $p = 2$ for the local spatial prediction task.

**PointNet**    For the *PointNet* network we use the same encoder architecture as for the VAE, but add a final $1 \times 1$ regressor to $K$ feature-maps corresponding to $K$ keypoints (Jakab et al., 2018) with a standard deviation of 0.1 for the Gaussian masks. We resize predictability maps to $84 \times 84$ and concatenate them channel-wise before feeding it to the PointNet.

#### A.2.2    Transporter Re-implementation

We re-implemented the *Transporter* model (Kulkarni et al., 2019) for our baseline method. We used an encoder network of 4 *Conv-BatchNorm-ReLU* layers. The feature extractor network $\Phi$ uses kernel sizes $[3, 3, 3, 3]$, filters $[16, 16, 32, 32]$ and strides $[1, 1, 2, 1]$. The *PointNet* $\Psi$ uses the same architecture but includes a final $1 \times 1$ regressor to $K$ feature-maps corresponding to $K$ keypoints. 2D co-ordinates are computed from these $K$ maps as described in Jakab et al. (2018). The *RefineNet* uses the transpose of $\Phi$ with $2\times$ bi-linear upsampling to undo striding. We used the Adam optimizer with a learning rate of 0.0002 with a decay rate of 0.9 every 30000 steps and batch size of 64. We trained for a 100 epochs with an early-stopping parameter of 5 to prevent over-fitting.

The same hyperparameters are used across all environments except for the number of keypoints.

### A.2.3 RL AGENT AND TRAINING

The convolutional keypoint encoder in the *Transporter* and PKey+CNN models consists of 4 *Conv-BatchNorm-ReLU* layers with kernel sizes $[3, 3, 3, 3]$, strides $[1, 1, 2, 1]$ and filters $[128, 128, 128, 128]$. The final convolution layer activation is flattened and passed through a *Dense* layer with 128 units and ReLU activation. The convolutional keypoint encoder receives the keypoint features obtained in the following manner: let $\Phi(x_t)$ a $[H, W, C]$ tensor be the convolutional features and $\mathcal{H}(x_t)$ a $[H, W, K]$ tensor be the keypoint masks for an input image $x_t$. Keypoint features are obtained by multiplying keypoint masks with convolutional features and superposing masked features of all the $K$ keypoints.

The MLP architecture used as the building block throughout the graph neural network consists of a 2-layers, each having 64 ReLU units. We use a similar MLP (but having linear output units) as the positional encoder network to produce learned positional embeddings given keypoint centers. Output node values from the *Decode* block of the GNN are concatenated and fed into a 2-layer MLP, with 128 ReLU units each before being input to the agent.

For training the RL agents, we use Polyak weight averaging scheme at every step (constant=0.005) to update the target Q-network. We use Adam optimizer (Kingma & Ba, 2015) with a learning rate of 0.0002 and clip gradients to maximum norm of 10. For $\epsilon$-greedy exploration we linearly anneal $\epsilon$ from 1 to 0.1 over the entire 100,000 steps (or 400,000 frames).

During training we checkpoint policies by running 10 episodes on a separate validation environment initialized with a seed different from the one used for training. We checkpoint policies based on their mean scores on this validation environment. For final evaluation (reporting test scores), we take the best checkpointed models from 3 different runs and run 10 evaluation episodes for each of the 3 policies on 10 previously unseen test environment seeds. We report the mean (and std. dev.) for the 3 different policies using the median evaluation run over the 10 test seeds. We emphasize that different environment seeds are used for training, validation (checkpointing), and testing. Further we do not add any noise in our policy used for evaluation episodes.

## B ADDITIONAL RESULTS

### B.1 ADDITIONAL VISUALIZATIONS

### B.1.1 NUMBER OF KEYPOINTS

On the *Frostbite* environment we modulate the number of keypoints produced by the *PointNet* while keeping all other hyperparameters same as described above in Appendix A (visualization shown in Figure 6).

### B.1.2 EFFECT OF LAYER CHOICE

On the *Space Invaders'* environment we modulate the feature layer(s) chosen $= \{(0), (0, 1), (2, 3), (0, 1, 2, 3)\}$ for the local spatial prediction task while keeping all other hyperparameters same as described above in Appendix A.

### B.1.3 SEEDS

We evaluate the stability of our keypoint discovery method over 5 random seeds on *Enduro* and *Battlezone* environments with all hyperparameter settings same as described in Appendix A and the qualitative results are shown in Figure 8 and Figure 9.

### B.1.4 KEYPOINT MASKED REGIONS

Figure 10 shows the keypoint mean (center) as well as Gaussian mask around it. We can see clearly from Figure 10 that although the keypoint centers might be slightly focused towards the borders of *predictability* their corresponding attention windows ensure good coverage of salient object parts in the scene.

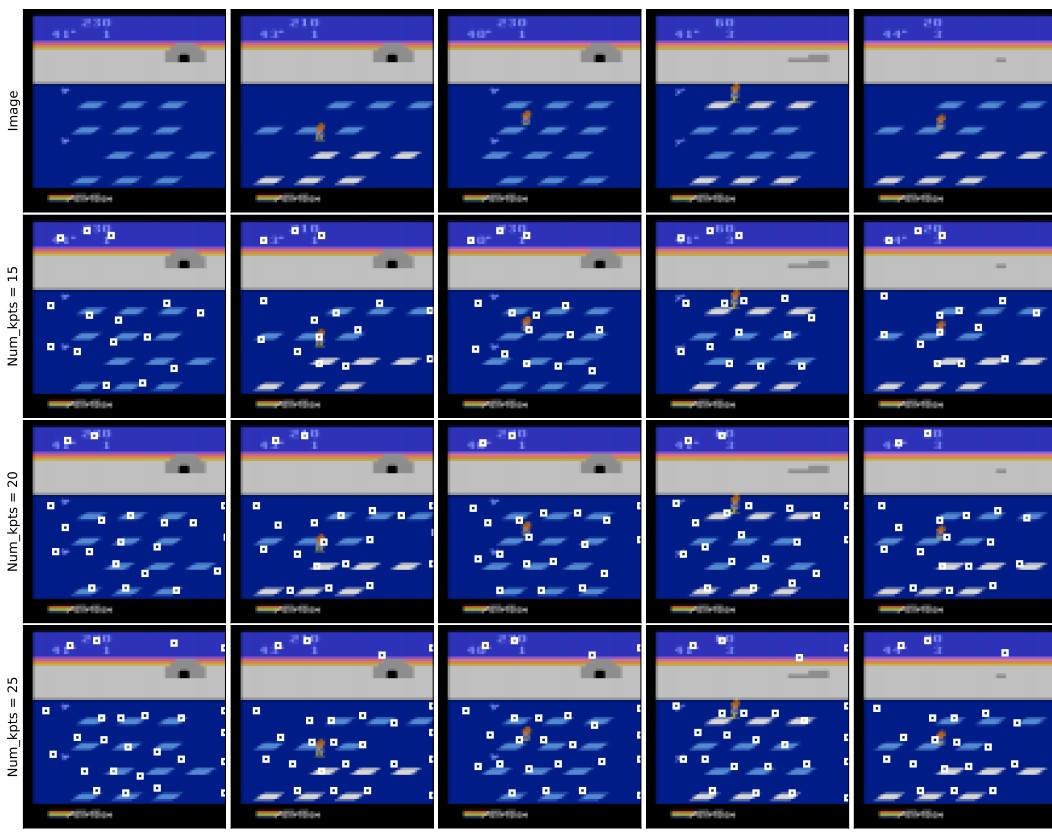

Figure 6: Image (row 1), keypoints (15, 20 and 25) overlaid on image (rows 2, 3 and 4). With 15 keypoints (row 2) we can see that our model prioritizes the keypoints to capture only the most salient object parts in the scene i.e. ice floats, player, scoreboard etc. As we increase the number of keypoints available it places the excess keypoints on the right border, after having captured all the highly salient object parts i.e. ice floats, player, birds etc.

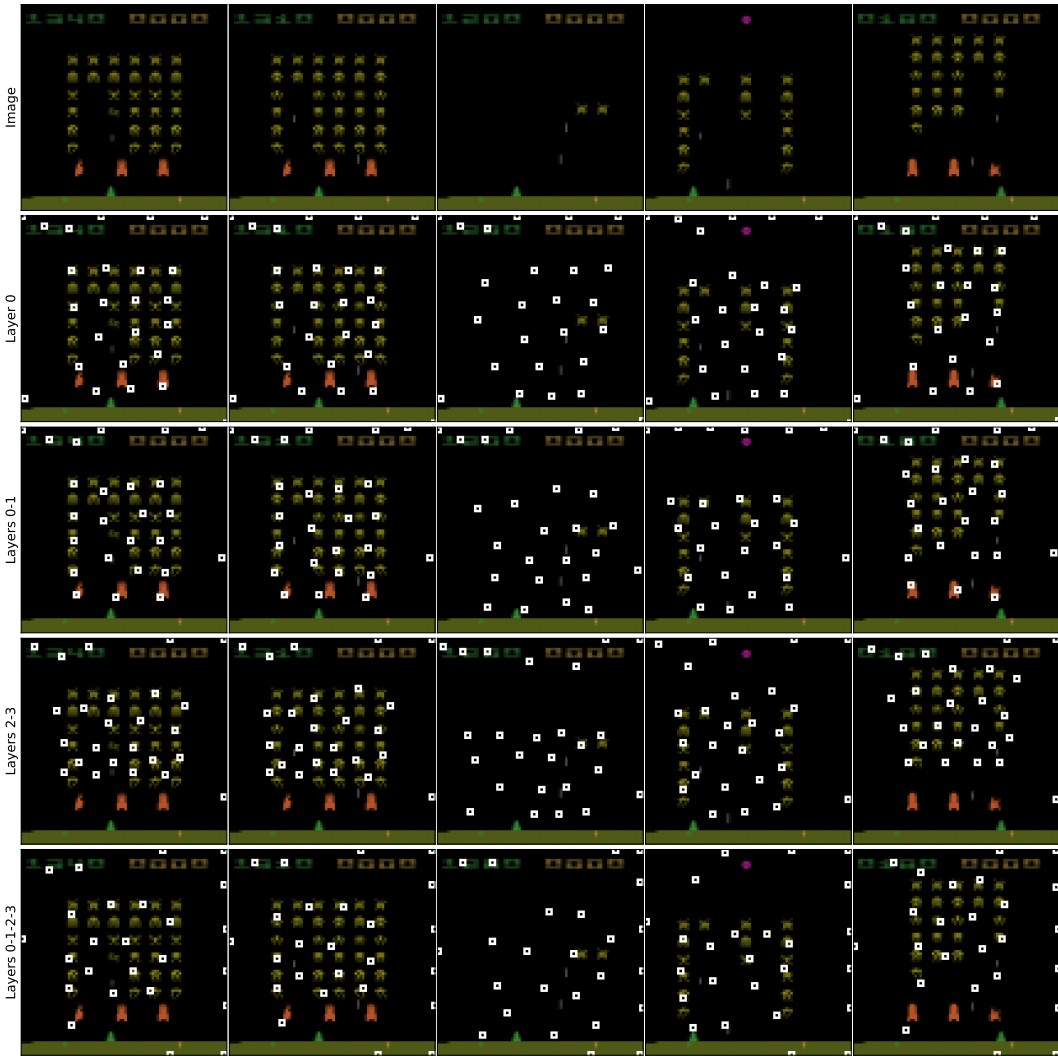

Figure 7: Image (row 1), keypoints using feature layer(s) = $\{(0), (0, 1), (2, 3), (0, 1, 2, 3)\}$ (rows 2, 3, 4 and 5 respectively) for local spatial predictability task. We can see that using feature layers 0 and 1 achieves the best keypoint quality. Including higher feature layers (rows 4 and 5) results in more uneven grouping of rows of enemy aliens to keypoints (columns 1, 2 and 6 row 5) and failure to track orange shields (rows 4 and 5).

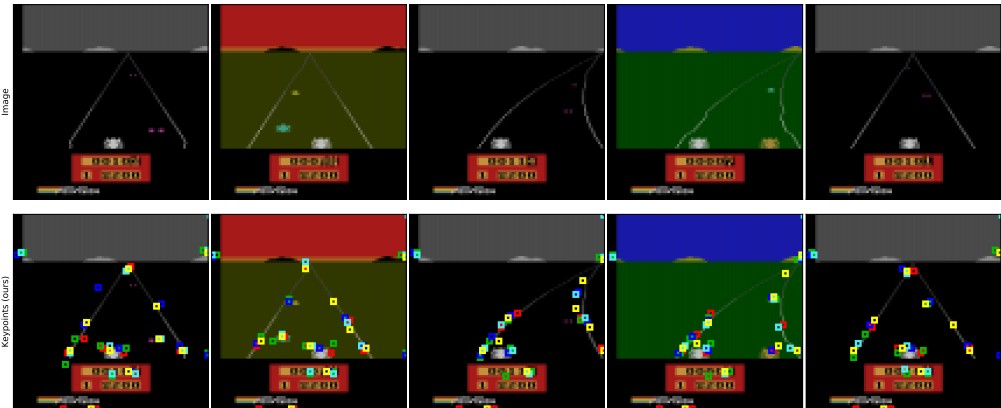

Figure 8: image (row 1), keypoints colour coded from 5 random seeds overlaid simultaneously on image (row 2). As we can see the learned keypoints remain stable across different runs and capture the salient object parts i.e. player, edges of the road, oncoming traffic, scoreboard etc. each time.

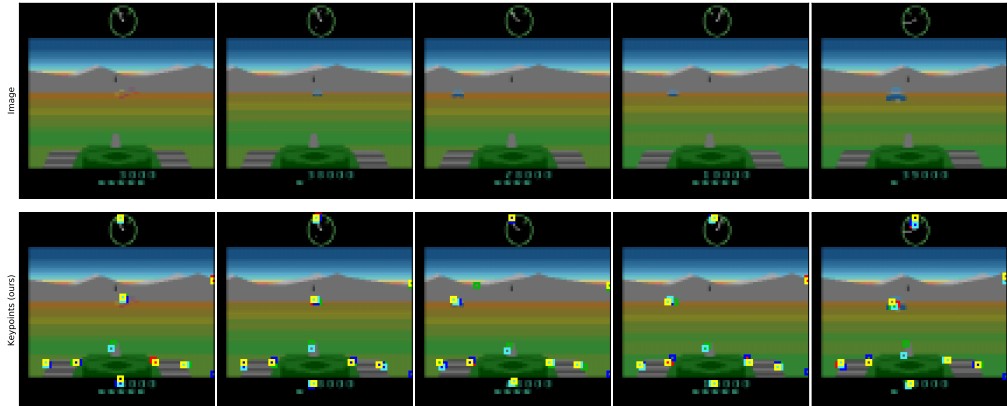

Figure 9: image (row 1), keypoints colour coded from 5 random seeds overlaid simultaneously on image (row 2). As we can see the learned keypoints remain stable across different runs and capture the salient object parts i.e. player tank parts, enemy tank, orientation dial at the top etc. each time.

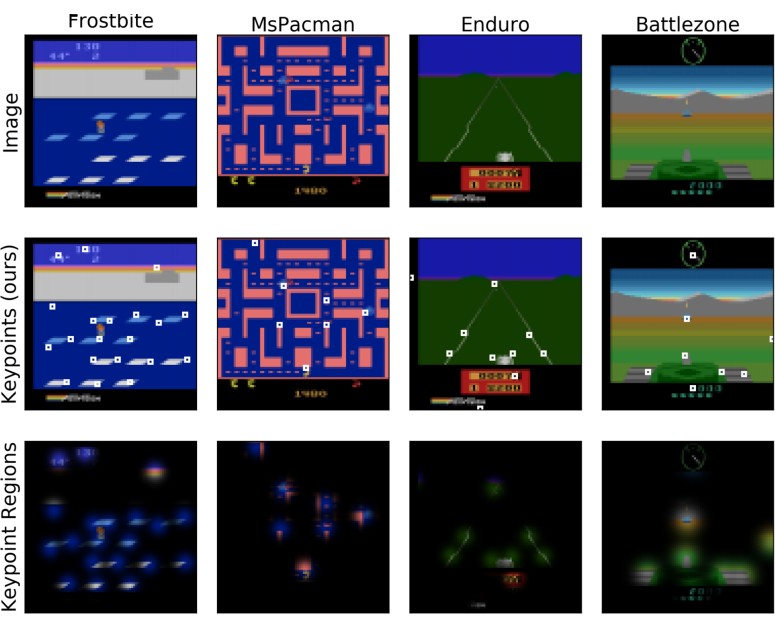

Figure 10: image (row 1), keypoint centers (row 2) and keypoint windows/masks (row 3).

