# OpenReview forum: "Unsupervised Object Keypoint Learning using Local Spatial Predictability"
_ICLR.cc/2021/Conference — ICLR 2021 Spotlight_

### Official Review · AnonReviewer4 · 2020-10-28
**Official Blind Review #4**

**Rating:** 9
**Confidence:** 3

**Review:**

The authors tackle the problem of self-supervised representation learning, and validate their approach on downstream Reinforcement Learning tasks. Building on the insight that predictability in local patches is a good inductive bias for salient regions that characterize objects, the authors propose a well-reasoned, well-engineered and thoroughly validated pipeline for learning object keypoints without supervision. The authors present a wide range of ablative studies to validate their design choices, and demonstrate the superiority of their method both illustratively as well as quantitatively on a number of standard Atari benchmarks.

The paper is very well written and clearly explained, with the assumptions clearly stated, and design choices thoroughly validated through ablative studies. Overall the authors make a very compelling argument for using local predictability as an intrinsic property of objects, leading me to recommend accepting this paper for publication.

Pros:
+ The intro motivates the problem well, contrasting the proposed method with a number of key recent methods. The implementation details are well recorded in the supplementary, with the added mention of releasing the source code
+ The keypoint detection pipeline is well reasoned and well explained: using the error map obtained through the spatial prediction task to recover keypoints via a bottleneck with limited capacity is a neat idea. The authors ablate a number of design choices (number of keypoints, which encoder layers to use); Figure 1 and 2 are great at showing the high-level components of the method as well as (intermediate) outputs
+ The comparison against Transporter is thorough and well analyzed. Fig2.b. provides a very clear insight into the limitations of Transporter, showing that the method proposed by the authors is able to achieve some robustness to visual distractors. PKey-CNN uses a similar method as Transporter for encoding keypoint features for downstream tasks, and thus serves to show that the keypoints identified are indeed superior. PKey-GNN further increases performance on a number of Atari games.
+ Very good ablative analysis and qualitative examples.

Some questions:
+ Do the authors have any further insights regarding why PKey-GNN would perform worse than PKey-CNN? While the authors’ reasoning makes sense, in my understanding a GNN based approach should be able to model any kind of interaction.
+ The authors demonstrate impressive results on a number of Atari games. I am wondering how this method would perform on a slightly more complex environment, i.e. CarRacing in OpenAI’s gym environment, or maybe even going as far as CARLA?
+ As I understand, PermaKey is first trained on Atari game rollouts, with the policy trained afterwards. Would it be possible to optimize both the keypoints and the policy together, end-to-end?

Post Rebuttal:
I thank the authors for their detailed and thorough response. All my questions and concerns were addressed and I appreciate the discussion on end-to-end learning as well as the “Transporter + GNN experiment”. I am happy to maintain my original rating and recommend acceptance.

---

> ### Author Response · Authors · 2020-11-17
> **Response to AnonReviewer4 part 1/2**
>
> Thank you for taking the time to review our work in detail and provide valuable feedback. We’re glad that you found the ideas presented interesting and were excited to see how you share our enthusiasm for this work based on the many positive comments. Below are our responses to your queries:
>
> *“Do the authors have any further insights regarding why PKey-GNN would perform worse than PKey-CNN?”*
>
> This is an interesting question, which we have wondered about as well. However, it is difficult to connect the behavior of using CNN or GNN to specific properties of the environment while avoiding “reading into it too much”. Currently we believe that the main reason for the observed difference is that the CNN encodes a strong spatial inductive bias that better reflects how the downstream task should be performed on some environments. Indeed, we agree that in principle the GNN is the more general approach and a more natural choice when treating keypoints as a set representation. In fact, the main motivation for considering a CNN-based approach for processing keypoints was to provide a fair comparison to Transporter, which contributed this design.
>
> We conducted an additional experiment (based on R1’s suggestions) where we combine our implementation of the Transporter with the GNN-encoder to see if we can observe similar fluctuations across environments (which would bring further evidence to our current hypothesis). The following table shows results of “Transporter + GNN” appended to Table 1 of the submission. Note, that the relative improvements of “Transporter + GNN” (row 2) compared to “Transporter + CNN” largely mirrors the relative changes shown by analogous PermaKey variants (CNN vs GNN keypoint encoder) across all environments.
>
> | Table. 1         | Transporter (ours) + CNN | Transporter (ours) + GNN| PermaKey + CNN    | PermaKey + GNN   |
>
> | battlezone   | N/A                                       | N/A                                        | 10266.7 +- (3272.1) | 12566.7 +- (3297.9) |
>
> | mspacman  | 983.0 +- (806.3)                   | 838.3 +- (537.3)                    | 1038.5 +- (417.1)     | 906.3 +- (382.2)       |
>
> | frostbite       | 263.7 +- (14.9)                     | 465.0 +- (442.1)                    | 310.7 +- (150.1)       | 657.3 +- (556.8)       |
>
> | seaquest      | 455.3 +- (160.6)                   | 296.0 +- (82.51)                    | 520.0 +- (159.9)       | 375.3 +- (75.4)         |
>
> *“how this method would perform on a slightly more complex environment, i.e. CarRacing in OpenAI’s gym environment, or maybe even going as far as CARLA?”*
>
> It would indeed be interesting to test the limits of our approach on more visually complex environments, such as highly realistic 3D driving-simulators like CARLA (we believe that OpenAI’s CarRacing environment is of comparable visual complexity as the Enduro environment on Atari for which we’ve shown results). However, with CARLA requiring working with complex visual scenes, high-resolution, 3D, heavy occlusion, etc. we can only speculate about how PermaKey will perform. Provided that a good spatial embedding can be learned (which may already pose a challenge with current VAEs) we expect that some interesting keypoints will be recovered, although it is likely that many others will be noisy due to the increased complexity of the local spatial prediction problem. Fortunately there may be several directions to improve this, for example by augmenting the local prediction problem to also consider the time-dimension to help with predicting information that is heavily occluded (but which moves together in a predictable manner). Similarly, using a spatial embedding that explicitly considers depth may help offset some of the complexity due to 3D.
>
> To summarize, we certainly do not expect PermaKey to work “out of the box” due to CARLA being a significantly more complex environment. However, we believe that the underlying idea of leveraging local predictability to learn salient object keypoints should also extend to this setting, provided that the prediction problem can be made tractable.
>
> ---- end of part 1/2

---

> > ### Author Response · Authors · 2020-11-17
> > **Response to AnonReviewer4 part 2/2**
> >
> > “Would it be possible to optimize both the keypoints and the policy together, end-to-end?”
> >
> > This is an interesting direction that we have so far not explored. Certainly in a game where new objects are introduced only at much later levels it would be beneficial to continue refining the keypoint module using the unsupervised loss as was also noted in [1]. However, to also let the current task affect the discovery of keypoints (eg. by backpropagating the policy gradient)  may not necessarily be beneficial. For example, a strength of purely unsupervised discovery is that keypoints are task-agnostic and equally apply when, for example, the rules of a game change. Similarly, this should in principle let us leverage observations from multiple different games to train PermaKey. On other other hand, we agree that by considering task-specific information the keypoints for a particular task may be easier to discover and the overall representation is expected to be less redundant (due to only focusing on what really matters for a particular game).
> >
> > Currently we believe that an approach based on top-down feedback will be able to incorporate some of these advantages. In that case, the agent is able to query (or put greater emphasis on) only the keypoints relevant for a particular task using a top-down attention module as in [4]. In that way an agent can focus only on the keypoints that matter for solving the given task, without sacrificing the generality of the discovered keypoints (i.e. them being task-agnostic). Such an approach, which we plan to study in future work, does not yet provide a way for the learning of keypoints to benefit from task-specific information, but rather limits the potential down-sides of a task-agnostic definition of saliency for keypoint discovery. Based on your comment will update the paper to include a discussion on end-to-end learning and potential future directions.
> >
> > [1] Kulkarni, Tejas D., et al. "Unsupervised learning of object keypoints for perception and control." Advances in neural information processing systems. 2019.
> >
> > [4] Mott, Alexander, et al. "Towards interpretable reinforcement learning using attention augmented agents." Advances in Neural Information Processing Systems 32 (2019): 12350-12359.
> >
> > ---- end of part 2/2 ---

---

### Official Review · AnonReviewer1 · 2020-10-28
**A very good paper! Very thorough experimentation.**

**Rating:** 7
**Confidence:** 4

**Review:**

The authors propose a novel approach to unsupervised key point detection based on predictability. The demonstrate their model on Atari tasks comparing to other key point detectors.

 Quality:

The authors compare their work both qualitatively and quantitatively to Transporter.

The authors show that their model picks out important key points that Transporter does not. Figure 5 is a great! It would also be good to show the distribution of predicted key points over multiple runs for other levels.

The authors train agents on Atari and compare their model to suitable baselines. It’s interesting that the GNN does not always outperform the CNN. This paper could be improved by also comparing to Transporter + GNN.

When constructing the error map, is this approach very sensitive to the receptive field and the number/ location of the neighbours? How would this approach handle larger / more complex objects?

Section 5.2 is interesting. What about if you have new objects that are not predictable, but are distractors? Would the model not create key points here? For example, adding in some smaller distractor shapes? Some randomly coloured dots? Or some missing pixels?

The ablation study is great!

This results section of this paper is very thorough and addressed a lot of the questions that came to mind when reading the introduction and methods section of this paper.

Clarity:

The authors argue that local predictability is an intrinsic property of an object with out giving more evidence for this. Perhaps the authors are hypothesising that this interpretation of objects will be more useful for downstream tasks? It’s not clear to me otherwise why this is an intrinsic property of an object?

The authors could improve their paper by being very clear about distinguishing focus on image regions that are unpredictable” from “local regions in input space that have high internal predictive structure” when describing objects and key points.

With the exception to the above, the introduction is well written and the methods section is easy to follow.

The model is designed to pick out points that are harder to predict, which is useful for ignoring background and finding the agent as nicely demonstrated in Figure 2 of Montezuma Revenge, but it’s not clear that this is a good definition of an object? For example the platforms in Frostbite may be very easy to predict, but you would need to know where they are in order to successfully navigate the environment. Also, it seems that predictability of a feature may depend heavily on the environment and any new object in an environment would be immediately picked out even if they are irrelevant to the task (i.e. new colours etc).

Could you explain better why you thing that the platforms in frostbite are assigned key points as they are?

Figure 3 is a really clear and nice example.

Originality and Significance:

This approach is novel and interesting and offers a new perspective on what an object can be and what definitions of an object may be useful for training agents.

Pros:
- Well written (with minor exception)
- Thorough results section.
- A novel approach to thinking about what an object is.
- Improvement over Transporter on Atari tasks.

Cons:
- Some confusion in the introduction about their definition of an object.
- There may be some limitation of the types (size and shape) of the objects that this model can assign key points to.
- There are some examples of objects being detected where it is not clear why those points are being detected according to the definition given in the paper. Explaining this more clearly would improve the paper.

---

> ### Author Response · Authors · 2020-11-17
> **Response to AnonReviewer1 part 1/3**
>
> Thank you for taking the time to review our work in detail and provide valuable feedback. We’re glad that you found the ideas presented interesting and were excited to see how you share our enthusiasm for this work based on the positive comments. Below are our responses to specific queries:
>
> *“Figure 5 is a great!. It would also be good to show the distribution of predicted key points over multiple runs for other levels.”*
>
> We will add similar plots showing distributions of predicted keypoints over multiple runs for 1-2 additional Atari environment(s) in the Appendix as part of the revision. We’re currently running these experiments and will notify you when this update has been made in the manuscript.
>
> *“This paper could be improved by also comparing to Transporter + GNN.”*
>
> Thank you for this suggestion. We agree that this would be a valuable experiment, since it may also help us better understand when GNNs or CNNs are preferred. Following table shows results of “Transporter + GNN” appended to Table 1 of the submission.
>
> | Table. 1         | Transporter (ours) + CNN | Transporter (ours) + GNN| PermaKey + CNN    | PermaKey + GNN   |
>
> | battlezone   | N/A                                       | N/A                                        | 10266.7 +- (3272.1) | 12566.7 +- (3297.9) |
>
> | mspacman  | 983.0 +- (806.3)                   | 838.3 +- (537.3)                    | 1038.5 +- (417.1)     | 906.3 +- (382.2)       |
>
> | frostbite       | 263.7 +- (14.9)                     | 465.0 +- (442.1)                    | 310.7 +- (150.1)       | 657.3 +- (556.8)       |
>
> | seaquest      | 455.3 +- (160.6)                   | 296.0 +- (82.51)                    | 520.0 +- (159.9)       | 375.3 +- (75.4)         |
>
> These results provide further evidence that the choice of GNN vs CNN is indeed environment specific due to each having different inductive biases, as we argued in the paper. Further it can be seen how PermaKey consistently outperforms Transporter, also when using GNNs for both.
>
> *“is this approach very sensitive to the receptive field and the number/ location of the neighbours? How would this approach handle larger / more complex objects?”*
>
> To be able to accommodate larger / more complex objects, the saliency (predictability) maps need to be computed at several scales (image resolutions) and integrated. Generally this adds only a small amount of overhead, since these maps correspond to different layers of the _same_ convolutional encoder. On Atari we experimented with several configurations of this and found that providing enough scales is most important (having too many, or having smaller / larger receptive fields only has a small impact). Notice, for example, how we are able to use the same configuration for these hyper-parameters across all environments to obtain good results.
> Further, we argue that through inspecting the inferred saliency maps it is easy to identify if enough scales have been provided. Finally, we note that many objects are hierarchical and can be decomposed into smaller / fine-grained structures that themselves consist of predictable information. Thus it is reasonable to expect that in many cases larger/more complex objects can be modelled as a collection of parts where each part is represented as a keypoint.
>
> --- end of part 1/3 ----

---

> > ### Author Response · Authors · 2020-11-17
> > **Response to AnonReviewer1 part 2/3**
> >
> > *“What about if you have new objects that are not predictable, but are distractors? Would the model not create key points here?”*
> >
> > Indeed, that is correct. Certain unpredictable distractors could result in wrongly placed keypoints when using PermaKey, although they will most likely also distract the Transporter (provided they alter --move-- between frames). In what situations such distractors will truly result in wrongly placed keypoints is a tricky issue, since the degree of predictability plays an important role in this case. For example although local pixel-level noise is not predictable, predicting the mean value may be used by PointNet as a reasonable approximation that lets it focus on more informative regions when placing its keypoints, i.e. to account for information that appears less regularly and frequently. It is also unclear how the learned spatial embedding by the VAE reacts to such local noise. Generally, we view our focus on information that is unpredictable from its surroundings as a prior for what is important in a visual scene. This frequently coincides with a focus on objects, which are known to be the foundation for solving many tasks. Nonetheless, we agree that the paper falls short at discussing the limitations of PermaKey in this regard, and we will update the conclusion to add a discussion containing these elements.
> >
> > *“It’s not clear to me otherwise why this is an intrinsic property of an object?”* *“Perhaps the authors are hypothesising that this interpretation of objects will be more useful for downstream tasks? It’s not clear to me otherwise why this is an intrinsic property of an object?”*
> >
> > It is difficult to arrive at a precise definition of an object. The answer to the question of “What is an object?” is still a topic of intense debate and discussion for researchers in the fields of cognitive science, psychology, AI, philosophy etc. Rather than deliberating on what is a complete definition of an object at a philosophical/meta-physical level, we instead opted for a functional definition of an object that emphasizes their benefits for representing complex visual information when solving real-world tasks.
> >
> > In particular, we define objects as abstract patterns in the visual input that serve as modular building blocks (i.e. they are self-contained and reusable independent of context) for solving a particular task, in the sense that they can be separately intervened upon or reasoned with. From this notion of an object, it follows that local predictability is an “intrinsic property”, unlike say “movement”, which is orthogonal to this. In that sense you are correct in that we have implicitly assumed that they are relevant for solving downstream tasks. On the other hand, we believe that our definition of a salient object as being regions of high-internal predictive structure is generally applicable and valid for a wide variety of tasks expected to be performed by a visually-situated agent. In the next revision we will clarify how “intrinsicness” follows from the notion of object that we have assumed and how this relates to the kind of downstream tasks that we are interested in solving. Thank you for commenting on this.
> >
> > *“it seems that predictability of a feature may depend heavily on the environment and any new object in an environment would be immediately picked out even if they are irrelevant to the task (i.e. new colours etc).”*
> >
> > PermaKey only attempts to model a bottom-up notion of saliency [3], where salient (pop-out) regions are determined purely from input stimulus signals, but without considering task-dependent (top-down) information. In that sense, by extracting object keypoints in a purely unsupervised manner we aim to remain task-agnostic to allow for greater re-use and generalizability. Provided that we keep on updating PermaKey as new information comes in this may result in a representation that is redundant or overcomplete as you pointed out (an irrelevant object would be an example of an “unpredictable distractor” following your previous point). However, we argue that this is not a disadvantage as this task-agnostic representation can later be adapted to a particular task, eg. by querying (or putting greater emphasis on) only the keypoints relevant for a particular task using a top-down attention module as in [4]. In fact, by having a redundant representation in this way it can readily facilitate other tasks, such as the “new object” acting as the key in Montezuma’s revenge. This would not require re-training the bottom-up vision module, which allows to maximally leverage unsupervised data collected across a variety of tasks. Although we have not explored this direction in the current submission, we are eager to begin experimenting with a more complete model of visual saliency that is capable of interacting with bottom-up keypoints in this way. We will highlight this property of PermaKey and direction for future research in the improved submission.
> >
> > --end of part 2/3

---

> > > ### Author Response · Authors · 2020-11-17
> > > **Response to AnonReviewer1 part 3/3**
> > >
> > > *“Could you explain better why you think that the platforms in frostbite are assigned key points as they are?”*
> > >
> > > Platforms seen at the default image resolution (layer-0 with conv receptive field of 4 pixels) might seem very predictable in it’s highly local context of other ice float parts. However, casting the same local prediction task of predicting the ice float features at layer-1 (image resolution 0.5X due to stride=2 in conv-encoder) given only blue background context would lead to some error spikes. This is due to the fact that at many locations in the image, it seems reasonable to expect blue background patches to surround other blue background patches. Only at certain locations white ice floats are located in the middle of blue background patches thereby driving some amount of error (surprise) in local predictability. In fact this phenomenon can be seen by comparing the error maps obtained at layer-0 and layer-1 (In Fig. 2(a) column 1, compare rows 4&5). Row 4 shows the error map computed at layer-0 while row 5 shows the error map at layer-1. Layer-1 has small error magnitude at ice float locations (Fig 2(a) column 1 row 5) due to this scenario where insufficient context has been provided to predict ice-floats. This is why it is important to perform the local spatial prediction task and aggregate error maps across several spatial resolutions and feature embeddings containing varying amounts of local vs global image information. This also highlights the role of integrating information from saliency maps computed across several resolutions as has been noted in the works of (Itti & Koch [5], Jagersand [6]) in order to compute final fixation (attention) maps. This allows us to accommodate objects of varying shapes, sizes and geometry etc.  We will try to clarify this better in the paper following your comments
> > >
> > > [3] Laurent Itti (2007) Visual salience. Scholarpedia, 2(9):3327.
> > >
> > > [4] Mott, Alexander, et al. "Towards interpretable reinforcement learning using attention augmented agents." Advances in Neural Information Processing Systems 32 (2019): 12350-12359.
> > >
> > > [5] Itti, Laurent, Christof Koch, and Ernst Niebur. "A model of saliency-based visual attention for rapid scene analysis." IEEE Transactions on pattern analysis and machine intelligence 20.11 (1998): 1254-1259.
> > >
> > > [6] Jagersand, Martin. "Saliency maps and attention selection in scale and spatial coordinates: An information theoretic approach." Proceedings of IEEE International Conference on Computer Vision. IEEE, 1995.
> > >
> > > ---- end of part 3/3

---

### Official Review · AnonReviewer2 · 2020-10-28
**Interesting idea**

**Rating:** 6
**Confidence:** 1

**Review:**

#### Summary

This paper works on unsupervised discovering keypoints in an Atari game frame to help improving Atari game performance. The keypoint discovery is based on predicting "predictable" local structure. I.e., the authors consider points that can not be predicted from its neighbor as good. Experiments show the learned keypoints performs better on 3 Atari games (Table. 1) than a counterpart keypoint discovery method, Transporter.

#### Strength

- The key idea of finding non-locally-predictable points as a representation of the game state is interesting, and specifically suitable for Games, where the backgrounds are mostly static and predictable.

- The technical implementation of the framework (Fig. 2) is clear and makes sense to me.

- Experimental results in Table. 1 is healthy. It shows the proposed method decently outperforms the Transporter counterparts.

#### Weaknesses

- The ablation studies are not exciting. These (number of keypoints/ spatial resolution of the embedding) are mostly design choice experiments and are better to be in the supplement materials. A more interesting ablation would be to quantitatively evaluate the quality of the points. Currently the paper only qualitatively shows the keypoint discovering results in Fig. 2 and claims an advantage to Transporter. This is not clear to the reviewer. The reviewer understands that there is no existing metric to evaluate keypoint discovery quality. However some proxy evaluation would also be helpful. For example, are the learned points temporally stable?

- As the key contribution is the keypoint discovery, it would be more convincing to compare with other unsupervised keypoint discovery methods besides Transporter if applicable. E.g., PointNet (Jakab et al 2018) that considers keypoints as a pixel-level reconstruction bottle-neck.


#### Summary

The paper proposed an interesting idea with reasonable results (better than a recent counterpart, Transporter). However, the reviewer does not have backgrounds in the specific experimental settings (Atari games), and can not assess the significance of the improvements. Comparisons with more keypoint discovery methods would make the results more convincing. My current rating is 6, but might change based on other reviews.

#### Post rebuttal

Thank you for providing the rebuttal. The rebuttal addressed my concern on comparing to other baselines. And it's fine to keep the design choice experiments in the main paper. However a proxy evaluation of key point evaluation is still missing and it will further strengthen the paper (I don't have a clear idea for the evaluation either). I keep my original rating of 6.

---

> ### Author Response · Authors · 2020-11-17
> **Response to AnonReviewer2**
>
> Thank you for taking the time to review our work in detail and provide valuable feedback. We’re glad that you found the ideas presented interesting. Below are our responses to your specific queries and general comments:
>
> *"and specifically suitable for Games, where the backgrounds are mostly static and predictable.”*
>
> A small clarification. We note that although static backgrounds are usually predictable and thereby accounted for by PermaKey, the same applies to moving backgrounds that contain predictable content, such as in Battlezone and Enduro (Fig. 2(a) columns 3&4). Here we observe that Transporter tends to bias keypoints to regions that alter between frames (due to its “motion bias”) and as a result struggles to capture the truly salient objects such as tank parts. In contrast, PermaKey is more robust to such “noise” by focusing on predictability and places keypoints that better reflect our own understanding of what is important in the scene.
>
> *“These (number of keypoints/ spatial resolution of the embedding) are mostly design choice experiments and are better to be in the supplement materials”*
>
> Thank you for the feedback. Our main motivation for discussing the findings from these ablations in the main text was to inform readers about important considerations (such as the spatial resolution) when applying PermaKey on new environments outside Atari, even though we were able to reuse the same configuration across different Atari games without problems. Similarly, for the number of keypoints, another potentially important hyper-parameter, we were able to demonstrate how PermaKey is not overly sensitive to this choice. Considering also the feedback from the other reviewers, we are currently leaning towards keeping these results in the main text.
>
> *“However some proxy evaluation would also be helpful. For example, are the learned points temporally stable?”*
>
> We are eager to quantitatively evaluate the learned keypoints, but the absence of a “ground-truth” makes this difficult as you also noted. The proxy that we have used thus far is to measure how well the learned keypoints are applicable for solving downstream tasks, i.e. for solving Atari games. We are happy to expand on this by measuring “temporal stability”, but we would like to ask for some clarification regarding how to measure this. We could perhaps extract keypoints for a sequence of consecutives frames and cluster them spatially to obtain an indication of how much keypoints move across frames (i.e. the diameter of the cluster). But then it will still be difficult to interpret this result in the absence of ground-truth information about how much the true salient regions move. Generally, based on the qualitative results, we expect temporal stability (as we currently understand it) not to be an issue, as is somewhat evident from the obtained scores on Atari, where the _recurrent_ controller requires aggregating information across multiple different frames (using our keypoints) to perform an action.
>
> *"It would be more convincing to compare with other unsupervised keypoint discovery methods besides Transporter if applicable. E.g., PointNet (Jakab et al 2018)”*
>
> We would like to point out that in the Transporter paper [1], it was already demonstrated both qualitatively (Figure 2 on page 3) and quantitatively (Figure 4 on page 6) that the Transporter keypoints are superior compared to those produced by PointNet [2] on the same Atari suite of environments. Therefore in comparing to the Transporter we already compare to the stronger of these two baselines. Since PermaKey was able to outperform the Transporter we argue that it is reasonable to expect that PermaKey would outperform PointNet even further.
>
> [1] Kulkarni, Tejas D., et al. "Unsupervised learning of object keypoints for perception and control." Advances in neural information processing systems. 2019.
>
> [2] Jakab, Tomas, et al. "Unsupervised learning of object landmarks through conditional image generation." Advances in neural information processing systems. 2018.

---

### Author Response · Authors · 2020-11-23
**General Response following paper revision**

We have updated the current submission to include many of the valuable suggestions and comments that were made by the reviewers. In particular, we have made the following changes in the revised submission:

1) We have clarified our assumptions regarding the nature of objects and how it can be argued that local predictability is an intrinsic property of such objects in this case.

2) We have added experimental results for 'Transporter+GNN' added in Table. 1. It can be seen how PermaKey outperforms Transporter also in this case, when considering a GNN as a mechanism for down-stream processing for both. Further, it can be observed how the benefits of using a GNN compared to a CNN depend on the considered environment. In environments where we previously observed that CNN outperforms GNN for PermaKey, we now observe the same trends when using Transporter. As we have previously argued, it suggests that the stronger spatial inductive bias in the CNN can be beneficial in some cases, even though the GNN offers a more general and principled approach.

3) We have highlighted the importance of considering feature embeddings that trade off local and global information and integrating their corresponding error maps to obtain keypoints at multiple different spatial resolutions. In particular, we now discuss the Frostbite ice floats example in the main text to clarify how we are able to accommodate objects of varying shapes, sizes and geometry in this way.

4) We have added a discussion (section 6), which discusses possible limitations of our approach with respect to discovering unpredictable distractors, such as task-irrelevant objects. It is discussed how PermaKey only attempts to model a bottom-up notion of visual saliency, which may in some cases yield a redundant or overcomplete representation with respect to solving a particular task. It is also discussed how this may not necessarily be disadvantageous due to the learned representation being better reusable and generalizable in this case. In particular, we now suggest mechanisms for top-down keypoint selection to refine the same representation across a number of tasks. Finally, point out potential benefits and limitations of unsupervised learning vs end-to-end learning of keypoints together with a policy.

5) We have added Figure 5 for an additional environment (Figure 9) in Appendix B, where it can be seen how the learned keypoints are stable also in this case.

---

### Decision · Program_Chairs · 2021-01-07
**Final Decision**

**Decision:**

Accept (Spotlight)

**Comment:**

Reviewers all agreed that this submission has an interesting new idea for learning object/keypoint representations: parts of a visual scene that are not easily predictable from their neighborhoods are good object candidates. Experimental gains on various Atari games are convincing. The main drawback at this point is that the evaluation is limited to visually rather simple settings, and it is unclear how the approach will scale to more realistic scenes.